# Exploring the Association between Delirium and Malnutrition in COVID-19 Survivors: A Geriatric Perspective

**DOI:** 10.3390/nu15224727

**Published:** 2023-11-09

**Authors:** Sarah Damanti, Marta Cilla, Giordano Vitali, Valeria Tiraferri, Chiara Pomaranzi, Giulia De Rubertis, Rebecca De Lorenzo, Giuseppe Di Lucca, Raffaella Scotti, Emanuela Messina, Raffaele Dell’Acqua, Monica Guffanti, Paola Cinque, Antonella Castagna, Patrizia Rovere-Querini, Moreno Tresoldi

**Affiliations:** 1Vita Salute University, 20100 Milan, Italy; tiraferri.valeria@hsr.it (V.T.); pomaranzi.chiara@hsr.it (C.P.); derubertis.giulia@hsr.it (G.D.R.); delorenzo.rebecca@hsr.it (R.D.L.); castagna.antonella@hsr.it (A.C.); rovere.patrizia@hsr.it (P.R.-Q.); 2General Medicine and Continuity of Care Unit, IRCCS San Raffaele Hospital, 20132 Milan, Italy; vitali.giordano@hsr.it; 3Center for Liver Disease, Division of Internal Medicine and Hepatology, IRCCS Ospedale San Raffaele, 20132 Milan, Italy; cilla.marta@hsr.it; 4Unit of General Medicine and Advanced Care, IRCCS Ospedale San Raffaele, 20132 Milan, Italy; dilucca.giuseppe@hsr.it (G.D.L.); scotti.raffaella@hsr.it (R.S.); tresoldi.moreno@hsr.it (M.T.); 5Infectious Diseases Unit, San Raffaele Scientific Institute, 20132 Milan, Italy; messina.emanuela@hsr.it (E.M.); guffanti.monica@hsr.it (M.G.); cinque.paola@hsr.it (P.C.); 6Division of Immunology, Transplantation & Infectious Diseases, Università Vita-Salute San Raffaele, IRCCS Ospedale San Raffaele, 20132 Milan, Italy

**Keywords:** delirium, malnutrition, frailty, COVID-19 survivors, geriatric syndromes

## Abstract

Older individuals face an elevated risk of developing geriatric syndromes when confronted with acute stressors like COVID-19. We assessed the connection between in-hospital delirium, malnutrition, and frailty in a cohort of COVID-19 survivors. Patients aged ≥65, hospitalized in a tertiary hospital in Milan for SARS-CoV-2 pneumonia, were enrolled and screened for in-hospital delirium with the 4 ‘A’s Test (4AT) performed twice daily (morning and evening) during hospital stay. Malnutrition was assessed with the malnutrition universal screening tool (MUST) at hospital admission and with the mini-nutritional assessment short-form (MNA-SF) one month after hospital discharge. Frailty was computed with the frailty index one month after hospital discharge. Fifty patients (median age 78.5, 56% male) were enrolled. At hospital admission, 10% were malnourished. The 13 patients (26%) who developed delirium were frailer (7 vs. 4), experienced a higher in-hospital mortality (5 vs. 3), and were more malnourished one month after discharge (3 of the 4 patients with delirium vs. 6 of the 28 patients without delirium who presented at follow up). The 4AT scores correlated with the MNA-SF scores (r = −0.55, *p* = 0.006) and frailty (r = 0.35, *p* = 0.001). Frailty also correlated with MUST (r = 0.3, *p* = 0.04), MNA-SF (r = −0.42, *p* = 0.02), and hospitalization length (r = 0.44, *p* = 0.001). Delirium, malnutrition, and frailty are correlated in COVID-19 survivors. Screening for these geriatric syndromes should be incorporated in routine clinical practice.

## 1. Introduction

Older people, especially the frailer ones, are at high risk of developing geriatric syndromes while facing acute stressors such as hospitalization [1]. Geriatric syndromes are multifactorial conditions underpinned by common risk factors [2]. Thus, they can occur simultaneously and their relationship can be bidirectional [3,4,5].

About 25% of people aged 65 years and older are malnourished or at risk of malnutrition [6]. This percentage rises up to 50% in an acute hospital setting [7,8,9].

Delirium is an acute decline of cognitive function underpinned by predisposing chronic conditions and triggered by precipitating events. Delirium can be considered as a marker of brain vulnerability and has been proposed to be a sort of vital parameter in older people [4,5]. One third of people aged 70 or older hospitalized in general medicine wards suffers from delirium. Delirium is present in half the cases on admission, whereas in the other half, it develops during their stay [10,11]. Despite being so frequent, delirium often goes underrecognized and underdiagnosed.

Since the brain has elevated metabolic requirements, inadequate nutritional supply impairs the brain’s ability to work properly and predisposes to delirium [12,13]. 

On the other hand, people with a hyper- or hypoactive cognitive status, or excessively sedated, are at risk of not eating properly [14,15]. Moreover, delirium can favor the development of a functional dysphagia [16,17], creating a dangerous vicious circle where malnutrition predisposes to delirium and vice versa [18].

It should be also taken into account that the acute illnesses that can cause delirium may also impair the patients’ ability to eat by the oral route. For example, patients admitted for respiratory diseases often manifest dyspnea and are supplied with oxygen using masks that interfere with food intake. Despite the elevated metabolic requirements, enteral or parenteral nutritional supply is seldom administered during the first days of hospital stay, and, indeed, hospitalized individuals experience long fasting periods [19]. 

Moreover, the focus of physicians on the treatment of the acute pathologic conditions responsible for the hospitalizations, the scarce awareness of the importance of malnutrition and delirium, and the lack of time and personnel to perform multidimensional evaluations lead to an underdiagnosis of these dangerous geriatric syndromes. As a consequence, there is a dramatic increase in patients’ morbidity and mortality related to potentially preventable conditions [1,4,20,21]. 

COVID-19 has been associated with both an elevated risk of developing delirium and malnutrition [22,23,24,25]. 

An acute alteration in mental status has been reported in about 20–30% of COVID-19 hospitalized patients, with percentages up to 60–70% in severely ill ones [26].

Malnutrition (defined according to the Global Leadership Initiative on Malnutrition (GLIM) criteria) has been described in 42% of COVID-19 hospitalized patients [27]. Other studies confirmed that more than one third of patients hospitalized in general medical wards for COVID-19 experienced a weight loss of ≥5% during the acute stage of the disease [24,28].

Malnutrition is also a common complaint in COVID-19 survivors. Two French studies which evaluated malnutrition in terms of percentage of weight loss or reduced body mass index (BMI) identified a prevalence of malnutrition of 47% [29] and 33% [30] one month after hospital discharge.

We aimed at assessing the link between in-hospital delirium and post-discharge malnutrition in a sample of COVID-19 survivors hospitalized for SARS-CoV-2 pneumonia during the third wave of the COVID-19 pandemic.

## 2. Material and Methods

This prospective observational study was part of the COVID-BioB study (NCT04318366) [31], whose original aim was to characterize patients hospitalized for SARS-CoV-2 pneumonia through the prospective collection of demographic, anthropometric, clinical, and laboratory data. The COVID-BioB protocol was approved by the San Raffaele University Hospital Ethics Committee (protocol no. 34/int/2020). A convenience sample size was used, due to the setting of the COVID-19 pandemic.

All patients hospitalized between January and May 2021, during the third wave of the COVID-19 pandemic, for SARS-CoV-2 pneumonia were invited to a dedicated out-patient clinic, for a follow-up visit one month after hospital discharge. The data presented in this study refer to a sample of older patients (aged ≥65 years), who were screened for the presence of delirium during hospital stay in COVID-19 medical wards.

The 4 ‘A’s Test (4AT) was administered twice daily (morning and evening) by trained emergency and internal medicine residents, and was used to screen delirium. The diagnosis of delirium was confirmed by the clinical evaluation of a senior geriatrician (SD). Patients with any 4AT score ≥ 4 during hospital stay and a clinical evaluation of a geriatrician consistent with a diagnosis of delirium were included in the delirium group. 

The 4AT scale has been widely validated for the screening of delirium [32]. It is composed of 4 items: alertness, abbreviated mental test-4, attention (i.e., spelling the months of the year backwards), and acute change or fluctuating course. The score ranges from 0 to 12, and a score ≥ 4 is suggestive for the presence of delirium, even if clinical judgment is required to confirm the diagnosis. 

At hospital admission, the patients were also screened for the presence of malnutrition by their attending physicians, though the administration of the malnutrition universal screening tool (MUST) questionnaire [33]. MUST is composed of three items: BMI computation, assessment of the percentage of weight loss in the 6 months preceding the administration of the questionnaire, and the presence of an acute illness. Each item is given a score ranging from 0 (lowest risk of malnutrition) to 2 (highest risk of malnutrition). The single item scores are summed up to have the total score: 0 is indicative of a low risk of malnutrition, 1 of a moderate risk of malnutrition, and ≥2 of an elevated risk of malnutrition.

Modified GLIM criteria [34] were also computed at hospital admission and one month after hospital discharge. Modifications were needed, because it was impossible to assess muscle mass during hospital stay. These modified criteria have been already used in a study on malnutrition in COVID-19 patients [28]. Appendix A illustrates the modified GLIM criteria used in this study. 

The length of hospital stay and in-hospital mortality were retrieved from the consultation of medical records.

Follow-up visits took place one month after hospital discharges, from March to June 2021. 

During the visits, multidimensional geriatric evaluations were performed. These included anamnesis (with a particular attention on the presence of persistent or de novo COVID-19 symptoms), physical examination, measurement of the anthropometric parameters (height, weight, BMI, and waist and calf circumference) and screening for malnutrition (through the mini-nutritional assessment short-form (MNA-SF) questionnaire [35]. Total MNA-SF scores range from 0 to 14, with two cut-offs: a score of 12 or greater denotes normal nutritional status, a score between 8 and 11 denotes a risk of malnutrition, and a score of 7 or less denotes overt malnutrition.

Moreover, we screened for the presence of sarcopenia though the Strength, Assistance with walking, Rising from a chair, Climbing stairs, and Falls (SARC-F) questionnaire [36] and we assessed muscle strength (though the hand grip test [37]), and muscle performance (through the Short Physical Performance Battery [38]). Pulmonary function was assessed through the 6 min walk test [39].

To synthetize the patients’ clinical complexity and vulnerability to stressors, we computed a frailty index (FI) according to the criteria described by Searl et al. [40] (Appendix A). The variables included in the FI computation were the following: comorbidities, laboratory values and clinical parameters (SpO_2_/FiO_2_) during hospital stay (taking into account the complexity and the severity of COVID-19 disease), social information (like being institutionalized), and the presence of polytherapy and of the anticholinergic burden of chronic therapies computed through the anticholinergic cognitive burden (ACB) scale. The ACB score captures the severity of the accumulative anticholinergic cognitive burden due to the total medications taken by older adults [41]. Each deficit included in the FI was scored 0 when absent, and 1 when present. Thirty variables were included in the computation of the FI, thus conferring it a sufficient robustness. The number of deficits recorded for each patient was summed to create the numerator of the FI, and then divided by the total number of possible deficits included in the computation of the FI. In cases of missing data, the FI was calculated using an adequately reduced denominator, excluding the items for those whose data were missing [42]. The participants having more than 20% of missing variables (>6 variables) were excluded from the computation of the FI [43]. The score of the FI ranges from 0 to 1, with lower levels identifying fitter individuals. A cut-off point of ≥0.25 was used to define ‘frail’ individuals [44].

Appendix A illustrates the evaluations performed at different time points.

## 3. Statistical Analyses

Descriptive statistics were used to illustrate the baseline characteristics of the study population. Continuous variables were presented as mean and standard deviation (SD), when normally distributed, or with median and interquartile range (IQR), when data had a skewed distribution. Dichotomous variables were presented as number (N) and percentage (%). Comparisons of the baseline characteristics between patients with in-hospital delirium and without in-hospital delirium were performed with the t test or U Mann–Whitney test for continuous variables, and with the chi-squared test for categorical variables. Spearman’s correlations were used to explore the association among 4AT scores during hospital stay, frailty index score, MUST, and MNA-SF one month after hospital discharge. Cohen’s kappa coefficients were used to compare the agreement among the different tools used to detect malnutrition.

All statistical analyses were performed with IBM Corp. Released 2017. IBM SPSS Statistics for Windows, Version 25.0. Armonk, NY, USA: IBM Corp.

## 4. Results

In this prospective observation study, we enrolled 50 older patients (median age 79, 56% males) hospitalized for SARS-CoV-2 pneumonia. No patient had received the SARS-CoV-2 vaccine before being infected because vaccination was not yet available for older people in that period. No patient received invasive mechanical ventilation but four patients (8%) were supported with non-invasive mechanical ventilation. A total of 30 patients (60%) were treated with steroids, 27 (54%) with heparin, and 15 (30%) with antiviral therapies. Nine patients (18%) received biological drugs and fourteen (28%) antibiotics.

Thirteen (26%) experienced delirium during hospital stay. These patients were frailer, (54% vs. 11%, *p* = 0.001) compared to patients who did not develop an acute brain failure. Table 1 illustrates the baseline characteristics of the sample. In particular, patients with and without delirium did not differ for the prevalence of comorbidities (except for the prevalence of solid tumors which was higher in the patients without delirium (27%) vs. the patients with delirium (0%), *p* = 0.046), polytherapy, or nutritional status before hospital admission. On the other hand, the patients who experienced delirium during hospitalization had a lower pre COVID-19 weight compared to the patients who did not have delirium (64 kg vs. 74 kg, *p* = 0.03). According to the MUST criteria, only five patients, 10% of the sample, were malnourished before hospital admission (two people who would experience delirium during hospitalization and three who would not).

By using the modified GLIM criteria, eight patients (16%) were classified as malnourished at hospital admission. Weight loss and the nadir of body weight during hospitalization were not statistically different in patients with and without delirium. The median weight loss during hospital stay was 4 kg. For 54% of the sample, the amount of weight loss was greater than 5% of their body weight before hospital admission. Liver enzymes (transaminases) at hospital admission were higher in the patients who experienced delirium compared to the patients who did not (median AST 53 vs. 33, *p* = 0.018; median ALT 55 vs. 27, *p* = 0.046).

Eight patients died during hospital stay. In-hospital mortality was higher in the patients with delirium compared to the patients without delirium (39% vs. 8%, *p* = 0.01).

Ten patients did not present to the follow-up visits (four in the delirium group and six in the no delirium group).

Table 2 illustrates the main characteristics of the sample at the follow-up visits. One month after hospital discharge, the patients had a median weight of 3 kg less than their usual weight and a median MNA-SF score indicative of a risk of malnutrition (8 points). Nine patients (28%) were malnourished according to the MNA-SF tool (score < 8 points), whereas, according to the modified GLIM criteria, twenty-nine patients (91%) were classified as malnourished.

The concordance among the different malnutrition screening tools was poor (between MUST and modified GLIM criteria: κ = 0.023, *p* = 0.87; between MNA-SF and modified GLIM criteria: κ = 0.003, *p* = 0.98).

The people who experienced delirium during hospitalization were more malnourished (MNA-SF ≤ 7) at the one-month follow-up visits compared to the people who did not have delirium (75% vs. 21%, *p* = 0.007). One month after hospital discharge, 7 patients (22%) were at risk of sarcopenia (SARC-F score ≥ 4) and 23 patients (72%) were pre-sarcopenic (i.e., they had a reduced grip strength). The prevalence of pre-sarcopenia and the risk of sarcopenia were not different among the patients who experienced delirium and those who did not experience it.

Table 3 shows the results of Spearman’s correlations among malnutrition before and after hospitalization (MUST and MNA-SF, respectively), frailty (FI), and delirium (4AT scores).

Malnutrition before hospital admission was significantly correlated with the length of hospital stay (ρ0.5, *p* < 0.001). On the other hand, the length of hospital stay was also correlated with malnutrition after hospital discharge (ρ−0.36, *p* < 0.048). We found a significant correlation among various 4AT scores during hospital stay and MNA-SF one month after hospital discharge (Table 3).

Frailty was significantly correlated with malnutrition before and after hospitalization, length of hospital stay, and delirium (Table 3).

## 5. Discussion

In this prospective observational study, we found that COVID-19 patients who experienced delirium during hospital stay were frailer, had a higher in-hospital mortality, and were more malnourished one month after hospital discharge. The length of hospital stay was significantly correlated with malnutrition before and after hospitalization. Delirium during hospitalization (detected through the 4AT scores) was significantly correlated with the MNA-SF scores one month after hospital discharge. Transaminases at hospital admission were higher in the patients who developed delirium during hospital stay compared to the patients who did not have delirium. Finally, we detected correlations among frailty, malnutrition (before and after hospitalization), length of hospital stay, and delirium.

Our results are in line with the findings of a recent Danish study, conducted by Rosted et al., among 612 old COVID-19 patients hospitalized in a geriatric unit [12]. In particular, the prevalence of delirium (26% in our study vs. 20% in Rosted et al.’s study) and of malnutrition and delirium (15% in both studies) are similar.

In the study by Rosted et al., malnutrition was defined as a BMI below 25 kg/m^2^ plus loss of appetite and weight during hospital stay, or a weight loss of more than 5 kg within three months prior to admission. If we compare the percentage of the patients who experienced a weight loss ≥ 5% during hospital stay in our study, (54%) and the patients defined as malnourished by Rosted et al. (57%), they are very similar.

Compared to the study by Pironi et al. [28], who assessed the risk of malnutrition in 268 patients hospitalized in different care settings (intermediate care units, sub-intensive care units, intensive care units, and rehabilitation units) in Bologna during the first wave of the COVID-19 pandemic, the median weight loss during the acute phase of COVID-19 was quite similar to the one detected in our study, (5.3 kg in the study by Pironi et al. vs. 4 kg in our study). The small difference may be due to the inclusion of patients hospitalized in intensive care units in the study by Pironi, where the amount of loss of body weight might have been greater due to the severity of the disease. Moreover, the prevalence of malnutrition (49%) during hospital stay detected by Pironi et al. was similar to the percentage of the patients who experienced a weight loss of ≥5% during hospitalization in our study (54%).

Compared to the previous data of our group (Di Filippo et al.), collected during the first wave of the COVID-19 pandemic [24], the prevalence of malnutrition (according to the MNA-SF score) one month after hospital discharge was higher in this study which was conducted during the third wave of the COVID-19 pandemic (18% in this study vs. 7% in the study by Di Filippo et al.). However, the median age of the patients (78.5 in the present study vs. 59 in the study by Di Filippo et al.) and the median length of hospitalization (28 days vs. 8 days in the study by Di Filippo et al.) were both higher in this study compared to the one by Di Filippo et al., and could have impacted on the prevalence of malnutrition.

We confirmed the link between delirium and malnutrition, as previously detected among SARS-CoV-2 patients, both in the hospital [12] and nursing home setting [45]. Indeed, a bidirectional relationship between these two geriatric syndromes has been widely demonstrated [3,4,5]. On the one hand, delirium impairs nutritional intake, favoring malnutrition, but on the other hand, malnutrition compromises the metabolic supply to the brain, increasing the risk of developing an acute brain failure. Moreover, the SARS-CoV-2 inflammation favors the passage of inflammatory cytokines through the blood–brain barrier, which can trigger delirium and foster a catabolic state, which concurs with malnutrition [46,47,48,49,50,51,52].

Frailty, being characterized by a reduction of physiological reserves, increases the risk of developing adverse outcomes [53,54] including delirium, malnutrition, and prolonged hospitalization, as detected in our study.

The prevalence of pre-sarcopenia one month after hospital discharge was high in this study (46%). Unfortunately, we do not have data referring to muscle function in the pre-infection period. Thus, we cannot be sure that these people were not pre-sarcopenic before SARS-CoV-2 infection. However, acute sarcopenia during COVID-19 is common [55,56], and our results are in line with another study by our group on COVID-19 survivors, which found a prevalence of pre-sarcopenia of 46% one month after hospital discharge [57].

The concordance among the different malnutrition definitions was poor. This is a problem already described in the literature [58,59], and, indeed, the prevalence of malnutrition is influenced by the criteria used to define it.

At hospital admission, eight patients (16% of the study sample) were malnourished, according to the GLIM criteria, and five individuals (10% of the study population) had an elevated risk of malnutrition, according to the MUST score. Indeed, for being classified at an elevated risk of malnutrition with the MUST tool, at least two criteria among low BMI (18.5 ≤ BMI ≤ 20 kg/m^2^) and weight loss (5–10%) have to be present. Alternatively, a severe reduction in BMI (<18.5 kg/m^2^) or weight (loss > 10% in the previous months), or the presence of an acute disease, with a reduction of nutritional intake, for at least five days, is required for being classified at high risk of malnutrition. Instead, according to the GLIM criteria, just a phenotypic criterium (non-volitional weight loss and age-adjusted low BMI) plus an etiologic criterium (reduced food intake/assimilation or disease burden/inflammation) are sufficient to diagnose malnutrition.

Our findings are in line with the literature. It has been demonstrated that the definitions of malnutrition, which combine the criteria low BMI and weight loss, detect a lower prevalence of malnutrition, compared to definitions which use either low BMI or weight loss [59]. Moreover, it should be considered that the GLIM definition of malnutrition uses an age-adjusted BMI cut-off, which takes into account the fact that, in older people, lower BMI are associated with a higher risk of negative health outcomes (including malnutrition). Therefore, the age-adjusted cut-off is higher than the non-age-adjusted cut-off, and more older individuals are classified as malnourished with the age-adjusted cut-off.

Moreover, one month after hospital discharge, the GLIM definition of malnutrition described a higher prevalence of malnutrition (91%) compared to the MNA-SF (28%).

However, all COVID-19 survivors had a recent acute disease (and thus a recent acute inflammatory burden), and most of them experienced a weight loss during the acute phase of the SARS-CoV-2 infection. MNA-SF is a more multidimensional tool, and was the only one which detected a difference in the prevalence of malnutrition in the patients who experienced delirium during hospital stay and those who did not. Therefore, even if there is no consensus on which tool is better to assess nutritional status in COVID-19 patients, in this population, MNA-SF was found to be the most suitable tool to study malnutrition in COVID-19 survivors.

We found that transaminases at hospital admission were higher in the patients who developed delirium compared to the patients who did not. A recent prospective, multicenter, cohort study by Zhu et al. [60], who enrolled 4589 patients hospitalized with SARS-CoV-2 infection, detected an association between aspartic transaminase/alanine transaminase ratio and delirium. SARS-CoV-2 infection can damage the liver through various mechanisms [61,62]. Alterations in the hepatic function can thus be correlated with the severity of the SARS-CoV-2 infection, but can also interfere with the metabolism of some chronic psychotropic drugs, thus predisposing to delirium. This study has the merit of having confirmed the association between two neglected geriatric syndromes, delirium and malnutrition, among COVID-19 patients. To the best of our knowledge, this is the first study describing a correlation among 4AT scores during hospital stay and MNA-SF scores one month after hospital discharge. Improving screening for delirium and malnutrition with an ad hoc questionnaire is highly recommended. Indeed, the lack of recognition of these conditions is not free of consequences. On the contrary, both delirium and malnutrition are associated with serious adverse clinical outcomes. Implementing their recognition through the routine application of validated screening tools, and treating them through multidisciplinary interventions would improve the well-being of hospitalized older patients and also the course of acute diseases which lead to hospital admissions [63,64].

Some limits of this study deserve to be mentioned. First, we should note the reduced sample size, due to the emergency care setting of the COVID-19 pandemic, and the monocentric nature of the study that limits the generalizability of our results. However, our findings are in line with previous European studies with wider sample sizes. Another potential limitation is the underreporting of pre-existing dementia, which can have been complicated by behavioral disorders, misinterpreted as delirium during hospitalization. Last but not least, the weight at hospital admission was self-reported, and this could have introduced a recall bias in our study.

## 6. Conclusions

In-hospital delirium and post-discharge malnutrition are correlated in COVID-19 survivors. Screening for these geriatric syndromes should become part of the routine clinical practice to improve patients’ well-being and potentially reduce the burden of adverse outcomes linked to these conditions.

## Figures and Tables

**Table 1 nutrients-15-04727-t001:** Main characteristics of the study population at hospital admission and during hospitalization.

	All	Delirium	No Delirium	*p*
	N = 50	N = 13	N = 37	
Age	79 (IQR 73–85)	82 (IQR 77–89)	78 (IQR 73–84)	0.08
Males	28 (56%)	8 (62%)	14 (38%)	0.20
Institutionalized	2 (4%)	1 (8%)	1 (3%)	0.46
Frailty index	0.2 (IQR 0.13–0.24)	0.27 (IQR 0.2–0.32)	0.17 (IQR 0.10–0.23)	0.003
Frail (FI ≥ 0.25)	11 (22%)	7 (54%)	4 (11%)	0.001
Hypertension	34 (68%)	11 (85%)	23 (63%)	0.18
Diabetes	13 (26%)	2 (15%)	11 (30%)	0.47
Dyslipidemia	18 (36%)	5 (39%)	13 (35%)	1
Coronary artery disease	9 (18%)	4 (31%)	5 (14%)	0.21
Atrial fibrillation	8 (16%)	2 (15%)	6 (16%)	1
Heart failure	4 (8%)	0 (0%)	4 (11%)	0.56
Previous pulmonary embolism/deep venous thrombosis	4 (8%)	0 (0%)	4 (11%)	0.56
Stroke/TIA	4 (8%)	0 (0%)	4 (11%)	0.56
Arthrosis	4 (8%)	1 (8%)	3 (8%)	1
Depression	5 (10%)	1 (8%)	4 (11%)	0.61
Hepatic disease	1 (2%)	0 (0%)	1 (3%)	1
Anemia	9 (18%)	1 (8%)	8 (22%)	0.41
Solid tumor	10 (20%)	0 (0%)	10 (27%)	0.046
Vaccination for SARS-CoV-2	0 (0%)	0 (0%)	0 (0%)	N.A.
Number of chronic drugs	4 (IQR 3–7)	5 (IQR 3–9)	4 (IQR 3–7)	0.36
Chronic ACB score	0 (IQR (0–1)	1 (IQR 0–3)	0 (IQR 0–1)	0.17
Polypharmacy	22 (44%)	7 (54%)	15 (41%)	0.52
Chronic psychoactive drugs	15 (30%)	6 (46%)	9 (24%)	0.17
Psychoactive drugs during hospitalization	19 (38%)	8 (62%)	11 (30%)	0.05
Reported weight before hospital admission (kg)	71 (IQR 55–81)	64 (IQR 44–71)	74 (IQR 60–83)	0.03
Height (cm)	165 (IQR 159–170)	165 (IQR 156–167)	164 (IQR 159–172)	0.35
BMI before hospital admission (kg/m^2^)	27 (IQR 23–30)	23 (IQR 20–28)	27 (IQR 24–30)	0.12
MUST at hospital admission				0.39
Low risk of malnutrition	28 (56%)	7 (53%)	21 (57%)	
Moderate risk of malnutrition	4 (8%)	2 (15%)	2 (5%)	
Elevated risk of malnutrition	5 (10%)	2 (15%)	3 (8%)	
GLIM malnutrition at hospital admission	8 (16%)	3 (23%)	5 (14%)	0.32
Weight variation during hospitalization (kg)	−4 (IQR −7–−3)	−4 (IQR −10–−1.5)	−4 (IQR −7–−3)	0.83
Weight loss ≥ 5%	27 (54%)	5 (39%)	22 (53%)	0.54
Nadir weight during hospitalization (kg)	69 (IQR 55–75)	62 (IQR 50–76)	70 (IQR 56–75)	0.43
White blood cells at hospital admission (cells/mm^3^)	7200 (IQR 4675–9625)	7150 (IQR 5050–9700)	7200 (IQR 4625–9425)	0.71
Lymphocytes at hospital admission (cells/mm^3^)	110 (IQR 775–1525)	1350 (IQR 725–1800)	1100 (IQR 725–1475)	0.50
Hemoglobin at hospital admission (gr/dL)	13 (IQR 12–14)	13 (IQR 10–14)	13 (IQR 12–15)	0.44
Platelets at hospital admission (10^3^ cells/mm^3^)	195 (IQR 143–279)	183 (IQR 155–226)	203 (IQR 138–333)	0.73
Glycemia at hospital admission (mg/dL)	99 (IQR 82–121)	114 (IQR 89–185)	95 (IQR 74–112)	0.05
Urea at hospital admission (mg/dL)	45 (IQR 35–57)	46 (IQR 40–62)	42 (IQR 33–58)	0.37
Creatinine at hospital admission (mg/dL)	0.98 (IQR 0.77–1.23)	0.92 (IQR 0.76–1.15)	1.01 (IQR 0.77–1.24)	0.73
Na at hospital admission (mmol/L)	140 (IQR 137–142)	139 (IQR 135–144)	140 (IQR 137–142)	0.71
K at hospital admission (mmol/L)	4.2 (IQR 3.7–4.6)	3.9 (IQR 3.6–4.3)	4.4 (IQR 3.7–4.7)	0.33
AST at hospital admission (U/L)	42 (IQR 25–53)	53 (IQR 35–94)	33 (IQR 24–51)	0.02
ALT at hospital admission (U/L)	30 (IQR 18–46)	55 (IQR 22–73)	27 (IQR 18–36)	0.046
AST/ALT	1.39 (IQR 1.08–1.69)	1.33 (IQR 1.08–1.74)	1.39 (IQR 1.06–1.69)	0.93
LDH at hospital admission (U/L)	333 (IQR 239–411)	337 (IQR 259–467)	316 (IQR 238–398)	0.59
CRP at hospital admission (mg/L)	35 (IQR 17.4–74.8)	32.4 (IQR 13.4–116.3)	35.3 (IQR 18.1–66.5)	0.82
Length of hospital stay (days)	28 (IQR 15–35)	31 (IQR 21.5–37)	25 (IQR 13.5–32)	0.14
Biologic therapies for COVID-19	9 (18%)	1 (8%)	8 (22%)	0.86
Antiviral drugs	15 (30%)	1 (8%)	14 (39%)	0.58
Steroid treatment	30 (60%)	3 (23%)	27 (73%)	0.74
Antibiotics	14	2 (15%)	12 (32%)	0.43
Heparin	27 (54%)	2 (15%)	25 (68%)	0.27
Non-invasive mechanical ventilation	4 (%)	0 (0%)	4 (19%)	0.48
Invasive mechanical ventilation	0 (0%)	0 (0%)	0 (0%)	N.A.
Death during COVID-19	8 (16%)	5 (39%)	3 (8%)	0.01

N.A.: Not applicable.

**Table 2 nutrients-15-04727-t002:** Main characteristics of the study population at the one-month follow-up visits.

	All	Delirium	No Delirium	*p*
	N = 32	N = 4	N = 28	
MNA-SF one month after hospital discharge	8 (IQR 7–10)	7 (IQR 5–9)	9 (IQR 8–10)	0.12
Malnourished (MNA-SF ≤ 7)	9 (18%)	3 (23%)	6 (16%)	0.007
GLIM malnutrition one month after hospital discharge	29 (58%)	5 (39%)	24 (65%)	0.75
SPPB	11 (IQR 7.5–12)	6 (IQR 0–12)	11 (IQR 9–12)	0.68
Hand Grip (kg)	19 (IQR 12–25)	14 (IQR 12–15)	20 (IQR 13–26)	0.32
Pre-sarcopenic	23 (46%)	3 (23%)	20 (54%)	0.35
SARC-F	2 (IQR 1–4)	2 (IQR 1–3)	2 (IQR 1–5)	0.81
Calf circumference (cm)	36 (IQR 33–39)	38 (IQR 35–41)	36 (IQR 33–39)	0.24
Waist circumference (cm)	99 (IQR 91–103)	99 (IQR 91–113)	98 (IQR 90–102)	0.69
Weight one month after hospital discharge (kg)	74 (IQR 61–79)	73 (IQR 63–80)	74 (IQR 59–78)	0.76
BMI one month after hospital discharge (kg/m^2^)	27 (IQR 25–29)	28 (IQR 23–33)	26 (IQR 25–28)	0.42
Δweight before hospitalization—one month after hospital discharge (kg)	−3 (IQR −4–−0)	−7 (IQR −10–−3)	−3 (IQR −4–0)	0.11
6 MWT (m)	420 (IQR 300–460)	400 (IQR 400–400)	430 (IQR 275–460)	0.87
6 MWT (%)	90 (IQR 63–98)	90 (IQR 90–90)	88.5 (IQR 58–98)	1

**Table 3 nutrients-15-04727-t003:** Spearman’s correlations among malnutrition, frailty, and delirium.

		MUST	MNA-SF	FI	Length of Stay
MUST	r	1.000	0.19	0.30	0.50
	*p*		0.30	0.04	<0.001
	N		31	50	50
MNA-SF	r	0.19	1.000	−0.42	−0.36
	*p*	0.30		0.02	0.048
	N	31		31	31
FI	r	0.30	−0.42	1,000	0.44
	*p*	0.04	0.02		0.001
	N	50	31		50
Length of stay	r	0.50	−0.36	0.44	1.000
	*p*	<0.001	0.048	0.001	
	N	50	31	50	
4AT(1)	r	0.23	−0.30	0.35	0.26
	*p*	0.11	0.10	0.02	0.07
	N	50	31	50	50
4AT(2)	r	0.24	−0.46	0.35	0.22
	*p*	0.11	0.01	0.01	0.14
	N	48	29	48	48
4AT(3)	r	0.22	−0.45	0.33	0.17
	*p*	0.14	0.02	0.03	0.26
	N	44	27	44	44
4AT(4)	r	0.28	−0.44	0.37	0.18
	*p*	0.07	0.02	0.01	0.24
	N	45	27	45	45
4AT(5)	r	0.27	−0.55	0.42	0.17
	*p*	0.70	0.006	0.006	0.29
	N	41	24	41	41
4AT(6)	r	0.24	−0.33	0.33	0.23
	*p*	0.17	0.14	0.06	0.19
	N	35	21	35	35
4AT(7)	r	0.18	−0.50	0.41	0.31
	*p*	0.34	0.03	0.02	0.09
	N	30	18	30	30
4AT(8)	r	0.20	−0.43	0.52	0.26
	*p*	0.33	0.11	0.007	0.20
	N	26	15	26	26
4AT(9)	r	0.19	−0.25	0.43	0.22
	*p*	0.38	0.39	0.03	0.30
	N	24	14	24	24
4AT(10)	r	0.23	−0.53	0.48	0.11
	*p*	0.32	0.07	0.03	0.63
	N	21	12	21	21
4AT(11)	r	0.23	−0.55	0.36	−0.11
	*p*	0.34	0.13	0.14	0.66
	N	19	9	19	19
4AT(12)	r	0.002	−0.54	0.14	−0.13
	*p*	0.99	0.18	0.61	0.65
	N	15	8	15	15
4AT(13)	r	0.20	−0.52	0.33	−0.22
	*p*	0.58	0.37	0.35	0.55
	N	10	5	10	10

## Data Availability

The data presented in this study are available on request from the corresponding author. The data are not publicly available due to privacy.

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
