# Peer review of "Exploring the Association between Delirium and Malnutrition in COVID-19 Survivors: A Geriatric Perspective"

_nutrients, 2023, doi:10.3390/nu15224727_

Round 1
Reviewer 1 Report
Comments and Suggestions for Authors
Interesting study but I have a major concern on data analysis (see below). Please update the results with the new analysis which may change findings and conclusions.
Abstract
Please indicate when 4AT, MNA-SF, FI, and MUST were measured.
Method
There are different cognitive, functional, and nutritional status assessment methods used in this study (4AT, MUST, GLIM, MNA-SF, SARC-F, SPPB, 6MWT, FI) and they were used at different time points. I think there should be a table or a figure to summarize and indicate which one was used at which time point.
It is unclear how subjects were categorized into delirium vs no delirium with 4AT administered multiple times during the stay at the hospital. If the subject scored >=4 for only once would that be sufficient to categorize them into the delirium group? Please clarify.
Results
Table 2 - Shouldn’t sample size be lower than Table 1 given ten subjects were missing at the follow up and eight subjects died at the hospital before the follow up? Please modify the table accordingly. Also, it appears that the statistical tests were performed with the denominators being n=13 and n=37. This is NOT correct because they include subjects that were dropouts. Please make sure the denominators are the new sample size, not n=13 vs n=37. The new analysis could change the findings and the conclusion.
Discussion
Please discuss why there is a low concordance among MUST, GLIM, and MNA-SF, and which one is the most credible in this group of population.
Comments on the Quality of English Language
Some proofreading and editing are still needed as many sentences were written with spoken language. For example, Line 142 “till” should be changed to “until”.
Author Response
We thank the Editor and the Reviewers for the possibility they gave us to better and improving our manuscript. The text was revised according to their suggestions
REVIEWER 1
Interesting study but I have a major concern on data analysis (see below). Please update the results with the new analysis which may change findings and conclusions.
Abstract
Please indicate when 4AT, MNA-SF, FI, and MUST were measured.
We thank the Reviewer for this comment. The text was revised accordingly.
Method
There are different cognitive, functional, and nutritional status assessment methods used in this study (4AT, MUST, GLIM, MNA-SF, SARC-F, SPPB, 6MWT, FI) and they were used at different time points. I think there should be a table or a figure to summarize and indicate which one was used at which time point.
We thank the Reviewer for this comment. Accordingly, we add a supplementary table (Table S3) to clarify the tools used the evaluate the patients and when they were applied.
It is unclear how subjects were categorized into delirium vs no delirium with 4AT administered multiple times during the stay at the hospital. If the subject scored >=4 for only once would that be sufficient to categorize them into the delirium group? Please clarify.
We thank the Reviewer for this comment. We clarified in the methods (line 122-123) that patients with any 4AT score ≥ 4 during hospital stay and a clinical evaluation of a geriatrician consistent with a diagnosis of delirium were included in the delirium group.
Results
Table 2 - Shouldn’t sample size be lower than Table 1 given ten subjects were missing at the follow up and eight subjects died at the hospital before the follow up? Please modify the table accordingly. Also, it appears that the statistical tests were performed with the denominators being n=13 and n=37. This is NOT correct because they include subjects that were dropouts. Please make sure the denominators are the new sample size, not n=13 vs n=37. The new analysis could change the findings and the conclusion.
We thank the Reviewer for this comment. Indeed, the sample size was lower at the follow-up visits. Eight patients died during hospital stay and ten patients did not present to the follow-up visits. So only 32 patients were evaluated one month after hospital discharge. Among them 4 had experienced delirium during hospital stay and 28 did not. Statistically analyses were performed with the correct denominator, it was only an error in reporting the sample of the table. It was corrected according to the Reviewer suggestion.
Discussion
Please discuss why there is a low concordance among MUST, GLIM, and MNA-SF, and which one is the most credible in this group of population.
We thank the Reviewer for this comment. This point was clarified in the discussion.
Comments on the Quality of English Language
Some proofreading and editing are still needed as many sentences were written with spoken language. For example, Line 142 “till” should be changed to “until”.
We thank the Reviewer for this comment. English was revised by a professional English speaker.
Reviewer 2 Report
Comments and Suggestions for Authors
Major comments
1. This study did not show laboratory data relating with frailty, malnutrition and the severity of COVID-19 (i.e., albumin, creatinine, lipid profiles, HbA1c, CRP, lymphocyte, monocyte levels). This reviewer would like to know the associations between those laboratory data and the delirium.
2. The information about treatment for COVID-19 is important to interpret these results. The presence or absence of vaccination, specific drugs for COVID-19, oxygen inhalation and/or mechanical ventilation?
Minor comments
1. The introduction section is redundant and that has too many paragraphs. Please revised it.Major comments
Author Response
We thank the Editor and the Reviewers for the possibility they gave us to better and improving our manuscript. The text was revised according to their suggestions.
Major comments
- This study did not show laboratory data relating with frailty, malnutrition and the severity of COVID-19 (i.e., albumin, creatinine, lipid profiles, HbA1c, CRP, lymphocyte, monocyte levels). This reviewer would like to know the associations between those laboratory data and the delirium.
We thank the Reviewer for this comment. Information on laboratory data was added in Table 1 and in the result section (lines 221-223). Moreover, we discussed the association between elevated transaminases at hospital admission and delirium in the discussion (lines 356-363)
- The information about treatment for COVID-19 is important to interpret these results. The presence or absence of vaccination, specific drugs for COVID-19, oxygen inhalation and/or mechanical ventilation?
We thank the Reviewer for this comment. Information on the vaccination status of the patients and their treatment was added in the result section (lines 196 -201) and in Table 1.
Minor comments
- The introduction section is redundant and that has too many paragraphs. Please revised it.
We thank the Reviewer for this comment. The introduction was revised and reduced accordingly to the Reviewer’s suggestion